# Evaluating Language Models for Efficient Code Generation

**Jiawei Liu**🖋️       **Songrun Xie**⛵      **Junhao Wang**⛵
**Yuxiang Wei**🖋️      **Yifeng Ding**🖋️      **Lingming Zhang**🖋️

University of Illinois Urbana-Champaign🖋️     Tongji University⛵
{jiawei6, lingming}@illinois.edu

## Abstract

We introduce Differential Performance Evaluation (DPE), a framework designed to *reliably* evaluate Large Language Models (LLMs) for efficient code generation. Traditional coding benchmarks often fail to provide reliable insights into code efficiency, due to their reliance on simplistic test inputs and the absence of effective compound metrics. DPE addresses these issues by focusing on efficiency-demanding programming tasks and establishing an insightful compound metric for performance evaluation. DPE operates in two phases: To curate efficiency datasets, it selects efficiency-demanding tasks from existing coding benchmarks and generates computationally expensive inputs to stress the efficiency of LLM solutions. To assess the code efficiency, DPE profiles the new solution and compares it globally against a set of reference solutions that exhibit distinct efficiency levels, where the matched level defines its efficiency score. As a proof of concept, we use DPE to create EVALPERF, a benchmark with 121 performance-challenging coding tasks.

Our comprehensive evaluation draws interesting findings on the efficiency impact of model sizes, instruction tuning, and prompting. For example, while the scaling law fails to account for code efficiency, general instruction tuning benefits both code correctness and efficiency. We also evaluate the evaluation by examining the effectiveness of DPE, showing that EVALPERF is reliable and convenient to use even across platforms.

## 1 Introduction

With the increasing usage (GitHub, 2023; Amazon Web Services, 2023) of Large Language Models (LLMs) for code generation, comprehensively evaluating these LLMs is crucial for finding the next advancements. As such, the functional correctness in code generation (Chen et al., 2021; Austin et al., 2021) has been well-studied, where given a coding instruction in natural language, LLMs produce solutions whose correctness is assessed through test execution.

While code correctness ensures the program performs its intended behaviors accurately, code efficiency is equally crucial for building high-quality software. With the massive

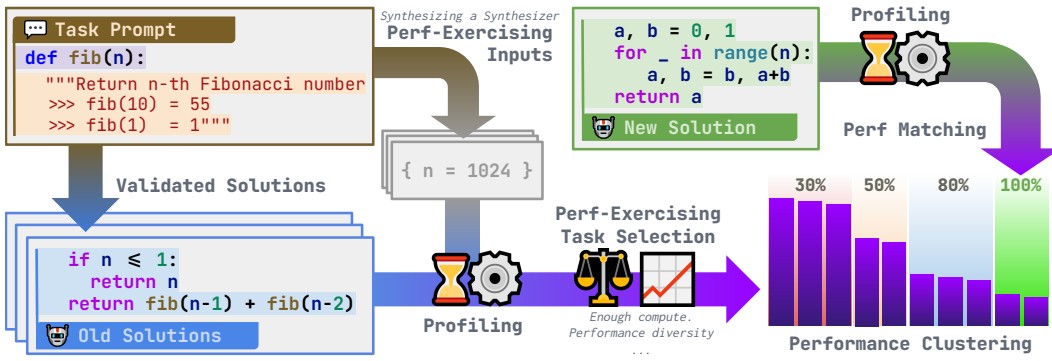

Figure 1: Overview of Differential Performance Evaluation

deployment of coding copilots, these assistants can help developers write low-latency, scalable, and cost-effective code by suggesting efficient algorithms, data structures, and coding patterns. More importantly, code execution can be a bottleneck for running the emerging Program-aided Language Models (Gao et al., 2023) (*e.g.,* GPT-4), motivating the generation of efficient code toward a smooth user experience.

Reasonably evaluating the code efficiency is important yet challenging. A naïve evaluation approach may simply record the execution runtime of validated solutions from existing benchmarks. However, such a strategy fails to provide reliable performance insights for two reasons:

**Limitation #1: Light computation.** Existing coding tasks commonly involve minimal computation, caused by small test inputs and simplistic control flow (*e.g.,* adding two numbers). However, it is not evaluation-friendly regarding code efficiency because lighter computation can incur larger result flakiness at orders of magnitude (Appendix A.1) due to its sensitivity to system noises. Meanwhile, code efficiency becomes less important at a tiny scale since all complexities are "equal" when $N$ is small. For example, a recursive implementation can be no slower than an efficient iterative solution when computing the first few Fibonacci numbers. Consequently, it is more meaningful to study code efficiency over large-scale data.

**Limitation #2: Inadequate metric.** Runtime speedup has been the de facto compound metric in the literature (Mendis et al., 2019; Zheng et al., 2020; Baghdadi et al., 2021) of efficiency optimization. While speedup is straightforward when studying a single optimization subject, averaging speedups over multiple tasks is confusing when interpreting the overall code efficiency of an LLM. For example, assuming model $A$ is slower than $B$ by $2\times$ on 99 tasks while outperforming $B$ on the only task by $100\times$, the average speedup says code from $A$ is generally faster than that of $B$ by $\frac{0.5 \times 99 + 100 \times 1}{100} = 1.495\times$, which may not align with general user perception. While we defer detailed discussions in Appendix A.2, the misperception comes from the huge scale variation of speedups across different tasks, calling for a more insightful compound metric for code efficiency.

To this end, we propose Differential Performance Evaluation (DPE), a general framework to curate performance-exercising programming challenges and perform effective code efficiency evaluation. From the *data* and *metric* perspective, DPE argues that effective performance benchmarking requires: *(i)* efficiency-challenging programming tasks to differentiate code solutions, and *(ii)* an unsightly metric to tell how far an LLM is to generate empirically optimal code. At dataset curation time, DPE takes a set of coding tasks as input and transforms them into tasks worth practicing for efficiency. Specifically, DPE generates performance-exercising test inputs for each task by *Synthesizing a Synthesizer* (SAS). SAS prompts an LLM with Chain-of-Thought (Wei et al., 2022) (CoT) few-shot learning to produce a *scale-controllable* test input sampler. Next, we tune the sampler to generate challenging yet computable inputs via exponential input sampling. Furthermore, we design filtering strategies to pick reliable tasks for performance evaluation. For each performance-exercising task, DPE samples a rich set of valid solutions and clusters them by performance characteristics. At evaluation time, DPE profiles a given new solution along with reference solutions and the ranking of the matched performance cluster determines its score. Below summarizes the contributions of this paper:

1. **Dimension:** While the correctness evaluation of code generation has been well studied, we deliver a new and important aspect to the community by studying the data curation and assessment for the efficiency evaluation of LLM-generated code.
2. **Technique:** We propose Differential Performance Evaluation (DPE) for effective efficiency evaluation. DPE curates performance-demanding coding tasks by sampling synthesized test input generators and using filters to ensure evaluator quality. A solution's efficiency is then globally assessed by referencing representative solutions.
3. **Benchmark:** Using DPE we create EVALPERF, including 121 performance-exercising programming tasks and test inputs. We also fully open-source and maintain the data curation pipeline and evaluator at `github.com/evalplus/evalplus` as part of EvalPlus.
4. **Study:** We extensively study the code efficiency of popular LLMs and draw interesting findings regarding the efficiency impact of model sizes, instruction tuning, and promptings. We also show that DPE can create inputs that are more performance-exercising than prior art by $4.8\times$ and EVALPERF can lead to consistent performance evaluation even across various platforms.

## 2   Differential Performance Evaluation

Figure 1 illustrates the overview of Differential Performance Evaluation (DPE), including *(i)* how to create a performance evaluation dataset to differentiate code performance (§§2.1 to 2.4); and *(ii)* how to evaluate new code solutions using the created dataset (§2.5).

At the high level, the input to the dataset creation phase is a set of programming tasks, *e.g.,* from HumanEval (Chen et al., 2021) and MBPP (Austin et al., 2021). As output, DPE produces a subset of performance-exercising tasks equipped with challenging test inputs. Specifically, we follow the steps below to transform and select performance-exercising tasks:

1. **Valid solution curation:** Given a programming task, we collect a rich set of correct solutions by sampling various LLMs and test execution.
2. **Performance-exercising input generation:** We maximize the performance difficulty of a coding task by synthesizing a test generator aiming to produce costly test inputs.
3. **Performance-exercising task selection:** We profile validated solutions using performance-exercising inputs and filter out tasks using various quality criteria.
4. **Performance clustering:** Based on the profiled performance, solutions for each task are partitioned into several clusters for performance reference at evaluation time.

During evaluation, if passing the correctness tests, the new solutions are profiled to compare against the reference solutions. Specifically, from slow to fast, the cumulative ratio of the cluster that includes the matched reference solution is the efficiency score of the evaluated solution. For example, in Figure 1 the new solution (the green box) matches the efficiency of the representative solution of the "100%" cluster, leading to a score of 100 (%) for this task.

### 2.1   Valid Solution Curation

To start with, DPE takes a set of programming tasks as inputs and assumes such tasks are equipped with task descriptions (*i.e.,* base prompts for the LLMs) and correctness tests. For example, these tasks can come from existing coding benchmarks such as HumanEval and MBPP. Next, we sample plausible solutions from diverse code LLMs and validate these solutions via test execution. Because correctness is the prerequisite for performance, we comprehensively validate plausible solutions using the rigorous tests from EvalPlus (Liu et al., 2023a).

### 2.2   Synthesizing a Synthesizer: Performance-Exercising Input Generation

While performance-exercising inputs are crucial, automating their creation can be challenging, as it is not always as simple as producing large integers. This is because these programming problems can define "large" inputs differently and require various structural and semantic constraints. For instance, when testing is_prime(n), a randomly large n often leads to a quick path of False since most numbers are divisible by smaller numbers (*e.g.,* half of the integers are divisible by two). Instead, the desirable test inputs are large *prime* numbers.

To this end, we propose *Synthesizing a Synthesizer* (SAS) to automatically produce performance-exercising inputs of different programming tasks by prompting powerful code LLMs to generate test generators. Further-

💬 **Instruction**
```
Generate function `perf_input_gen(scale: int)` to produce a "large"
input to exercise the efficiency of the `prime_num` function:
```
🔑 **Ground-truth Solution**
```
"""Write a function to check if a number is prime"""
import math
def prime_num(num):
    if num < 2: return False
    for i in range(2, math.isqrt(num)):
        if num % i == 0: return False
    return True
```
😕 **Chain of Thoughts**
```
Analysis:
1. Input format: An integer `n`
2. Time complexity: O(n)
3. Space complexity: O(1)
4. What kind of input can exercise its perf? Large prime numbers
```
⚒ **Input Generator**
```
# can reuse the `prime_num` function
# larger `scale` means larger input
# use case: prime_num(*perf_input_gen(scale))
def perf_input_gen(scale: int):
    for i in range(scale, 2, -1):
        if prime_num(i):  return (i,)
    return (2,)
```

Figure 2: Exemplifying *Synthesizing a Synthesizer*.

more, the generator function is controllable through a scale factor, allowing for tuning the complexity of generated test inputs.

**Generating input generator.** SAS applies few-shot prompting with Chain of Thoughts (CoT) for input generator synthesis, exemplified in Figure 2. Specifically, the end goal of prompting is to obtain the generator function (*i.e.,* perf_input_gen) illustrated in the "Input Generator" block (at the bottom). The generator function takes a scale factor as input and outputs performance-exercising test inputs according to the scale. The generator is expected to respect monotonicity over the scale factor, *i.e.,* a larger scale factor should lead the generator to produce a more challenging input. The context for generating the generator includes three parts: *(i)* an instruction clarifying the goal of code generation; *(ii)* a ground-truth solution helping the LLM understand the overall semantic and complexity; and *(iii)* a few question-answer pairs to activate CoT reasoning of the task complexity. By initializing the prompt using such few-shot samples, we further load the "instruction" and "ground-truth solution" block for a programming task under test synthesis and let a generative LLM follow the few-shot demonstration and produce a test input sampler.

**Exponential input sampling.** For each coding task, we sample performance-exercising inputs by running the generator function (*i.e.,* perf_input_gen) using different scale factors (*i.e.,* scale). Specifically, we start by setting the scale factor to 1 and sample test inputs by doubling the factor round by round. The sampled test inputs are evaluated through execution, and we stop generation when an input hits a time or memory limit on any validated solution (§2.1). By expanding the scale factor exponentially, we obtain the most performance-exercising input within our computational limits. Meanwhile, we use test execution to drop ill-formed generators and retry another sample of input generators at failure.

**Insight.** Prior work (Liu et al., 2023a; Li et al., 2022) also prompts LLMs to generate test inputs directly. However, such an approach is inapplicable for generating challenging inputs whose text representation can be huge, blowing up the context limits of LLMs. Meanwhile, it is also hard for LLMs to strictly follow the structural requirements and diversify the inputs during long-context generation. Hence, we highlight the indirect generation of complex test inputs via input generator synthesis. Recently, Zhang et al. (2023) proposed ALGO which uses ChatGPT Code Interpreter to create input generators for exhaustive validation. Different than ALGO, SAS aims for *performance-exercising* inputs via few-shot CoT and scale tuning, which does not rely on the powerful ChatGPT Code Interpreter.

## 2.3 Performance-Exercising Task Selection

Even with the most challenging test input, a programming task might not meet the requirement of performance diversity and runtime variation (*e.g.,* add two numbers). Consequently, we propose filtering strategies to drop undesired programming tasks. For every programming task, we profile all valid solutions $\{s_1, s_2, \cdots, s_n\}$ curated in §2.1 multiple times. Therefore, for solution $s_i$ we profile its execution for $k$ times and obtain a list of execution time[1] $T_i = [t_1, \cdots, t_k]$. As such, a selected programming task must meet the following criteria:

1. **Sufficient computation:** A performance-exercising task must experience a reasonably long execution. As such, we require $\min\{\text{mean}(T_i) | i \in [1, n]\} > t_{thresh}$, meaning that the execution of any solution must run longer than $t_{thresh}$.
2. **Low performance variation:** We require $P_{99}\{\text{CV}(T_i) | i \in [1, n]\} < CV_{thresh}$, where $\text{CV}(T_i) = \text{std}(T_i)/\text{mean}(T_i)$, *i.e., coefficient of variation*, and $P_{99}$ is the 99% largest variation.
3. **Performance diversity:** We apply a clustering method (to be discussed in §2.4) to adaptively cluster the solutions into several groups at different levels of efficiency. Therefore, we require the number of output clusters to be greater than $K$ where $K > 1$.

## 2.4 Adaptive Performance Clustering

Given a performance-exercising task, we cluster its reference solutions by their performance characteristics and use them to differentiate new solutions at evaluation time.

---

[1]For clarity, in this section, we use "execution time" or "runtime" to refer to the execution cost, which in practice can be generalized to other metrics such as the number of instructions.

```
1: def thresh(t): # smaller runtime ⇒ higher variation, needs larger thresh
2:   return BIAS + math.sqrt(WEIGHT / t)
3:
4: def cluster1d(time1d: List[float]) → List[List[float]]:
5:   time1d = np.sort(time1d)[::-1]              # slow to fast
6:   rdiff = -np.diff(time1d) / time1d[:-1]      # relative drop in 0-100%
7:   splitters = [i+1 for i, r in enumerate(rdiff) if r > thresh(time1d[i])]
8:   return np.split(time1d, splitters)          # return a list of clusters
```

Figure 3: The algorithm to adaptively segment solutions for each task based on their efficiency.

Figure 3 elucidates our adaptive clustering algorithm. Commencing from Line 4, the clustering algorithm takes a list of *mean* execution time (*i.e.,* 1-dimension) as input and dynamically partitions them into clusters based on their relative scale. Subsequently, we sort the execution time from slow to fast in Line 5. Denoting the sorted `time1d` as $\bar{T} = [\bar{t}_1, \bar{t}_2, \cdots, \bar{t}_k]$, in Line 6, we compute the relative difference in percentage, *i.e.,* $\delta\bar{t}_i = \frac{\bar{t}_i - \bar{t}_{i+1}}{\bar{t}_i}$. Following this, in Line 7, the algorithm employs an adaptive thresholding mechanism to determine the segmentation points over the sorted $\bar{T}$. Considering Figure 6, which indicates that smaller executions often exhibit larger variations, Line 7 calls the adaptive thresholding function in Line 1 which produces larger thresholds for smaller runtime. Specifically, to build a segmentation point at $t_i$ (left inclusive), the algorithm mandates that $\delta\bar{t}_i > bias + \sqrt{w/\bar{t}_i}$, where $bias$ and $w$ are hyper-parameters of the threshold minima and scale, respectively. Therefore, in Line 8, the algorithm returns the runtime clusters for each task by splitting them over the provided segmentation points.

## 2.5 Efficiency Scoring by References

In the final stage of dataset curation, we retain the slowest solution per cluster along with its cumulative percentage in each task. These solutions serve as benchmarks for efficiency scoring. During evaluation, both the reference solutions and the new solution are profiled in the same environment for performance assessment. Notably, we opt for a single representative solution per cluster as profiling all solutions during evaluation would be extremely time-consuming.

**Differential Performance Score.** For each task, we denote the reference solutions from the $m$ clusters as $[s_1, s_2, \cdots, s_m]$ and their corresponding cumulative ratios $[r_1, r_2, \cdots, r_m]$ ($r_1 > 0$ and $r_m = 100\%$). Meanwhile, we profile each reference point multiple times and obtain its mean runtime $\bar{t}_i$. Given a new and validated solution $s^*$ to evaluate, we use the same profiling procedure and obtain a mean runtime of $\bar{t}^*$. As such, we compute a *Differential Performance Score* (DPS) for $s^*$ as $\max\left(\{0\} \cup \{r_i \mid \bar{t}_i > \bar{t}^*\}_{i\in[1,m]}\right)$, *i.e.,* the cumulative ratio of the reference that is immediately slower than $s^*$. In addition, we also provide a normalized version of DPS by ablating the volume of solutions at each level, *i.e.,* $DPS_{\text{norm}} = \max\left(\{0\} \cup \{\frac{i}{m} \mid \bar{t}_i > \bar{t}^*\}_{i\in[1,m]}\right)$. The above discusses how to compute the performance score for one coding task, to compute the dataset-wise performance score we simply average over all passed tasks.

**Exemplification.** For example, a DPS of 80% implies that overall the LLM generates code whose efficiency can match or improve 80% of all LLM-generated solutions. Similarly, a $DPS_{\text{norm}}$ of 80% indicates that the code efficiency of the LLM can overall match or improve 80% of performance clusters, by ignoring the size of each cluster.

Lastly, DPS is not free as it requires to collect reference solutions of diverse efficiency levels.

# 3 EVALPERF: A Benchmark for Code Efficiency Evaluation

Based on Differential Performance Evaluation, we build EVALPERF, a new dataset with 121 performance-exercising programming tasks, for effective evaluation of code efficiency.

We put HumanEval+ (164 tasks) and MBPP+ (399 tasks) together as the initial tasks to DPE, given their more rigorous tests (Liu et al., 2023a) to safeguard correctness. To echo §2.1, we sample and test code solutions from 21 open LLMs that achieve over a pass@1 score of 50 on the EvalPlus leaderboard (Xie et al., 2024), where we sample 50 solutions for each model at a temperature of 0.8 for diversified generation. Next, we generate test input samplers (§2.2) using two few-shot samples and we start the generation at the "Chain of Thoughts" block in Figure 2 right after providing the ground-truth solution. Specifically, we use AWQ-quantized (Lin et al., 2024) DeepSeekCoder-instruct-33B (DeepSeek AI, 2023) as the generative LLM to produce 16 input generator samples for each task at a temperature of 0.8. We then sample concrete test inputs from these generators, starting with a scale factor of $2^1$ and increasing it exponentially until hitting the 20-second time wall or the 16GB memory wall. Of course, some generators could be broken and we filter them out via the running itself and its generated tests.

Task selection and clustering require profiling of these solutions. Specifically, we use the number of executed assembly instructions as the profile metric. This profile can be easily and natively obtained by querying the Performance Monitoring Units (PMU) of modern CPUs through system calls, such as perf_event (Weaver, 2013a) in Linux, which is pervasively available on most platforms. Compared to physical runtime, the #instruction is much more stable, resulting in negligible variation. Compared to software profilers such as architecture simulators, hardware counters provide low overhead (Wikipedia, 2024) and are easy to use through simple system calls. As such, we profile the #instructions for each solution over the performance-exercising input and filter the tasks using the criteria in §2.3. Specifically, we filter out tasks whose solutions can finish in $t_{thresh} = 10k$ instructions which is the scale of instructions for printing "hello world". We omit the variation criterion as we use the hardware performance counters for cost measurement. For clustering (§2.4), we set the base threshold (*i.e.*, *bias*) as 20% and the weight (*i.e.*, $w$) as 10k instructions for the adaptive threshold function in Figure 3, and for diversity require each task to have at least $K = 4$ performance clusters. As such, we build EVALPERF, a dataset with 121 performance-exercising coding tasks, equipped with computationally challenging inputs and solutions for performance reference.

Lastly, our future efforts will continuously extend EVALPERF using more coding tasks.

# 4 Evaluation

In §4.1 we study the code efficiency of recent code LLMs on EVALPERF and in §4.2 we evaluate the effectiveness of Differential Performance Evaluation.

## 4.1 Evaluating Code Efficiency

**Setup.** Following recent work (Lozhkov et al., 2024), we evaluate the performance as well as the correctness of programs synthesized by a series of open model families, including CODEL-LAMA (Rozière et al., 2023), DeepSeekCoder (DeepSeek AI, 2023), StarCoder (Li et al., 2023), and StarCoder2 (Lozhkov et al., 2024). For proprietary models, we evaluate GPT-4 Turbo (OpenAI, 2023) (*i.e.*, gpt-4-0125-preview) which is to date the leading model on the EvalPlus leaderboard (Xie et al., 2024). Specifically, we use up to four variants for each model type:

1. base: the base pre-trained model without instruction tuning.
2. instruct: the instruction-tuned model using its specialized instruction format.
3. perf-instruct: the instruction-tuned model using a prompt asking the model to "solve the programming task efficiently by writing a fast implementation".
4. perf-CoT: the instruction-tuned model with a zero-shot chain-of-thought prompt (Kojima et al., 2022) by adding "Think step by step" before the perf-instruct prompt.

By default, we generate 50 samples at a temperature of 0.2 for each programming task following Lozhkov et al. (2024). Yet, for cost mitigation, we limit it to 10 samples for GPT-4 Turbo. This approach is justified by the GPT-4 Turbo's higher accuracy in generating correct code. For correctness evaluation, we compute a comprehensive pass@1 on the sum of 164 HumanEval+ tasks and 399 MBPP+ tasks. For code efficiency analysis, for each task, we compute the average DPS and DPS$_{norm}$ values for the first 10 correct solutions in the 50 samples and report the score averaged across all tasks. Notably, we compare models in a

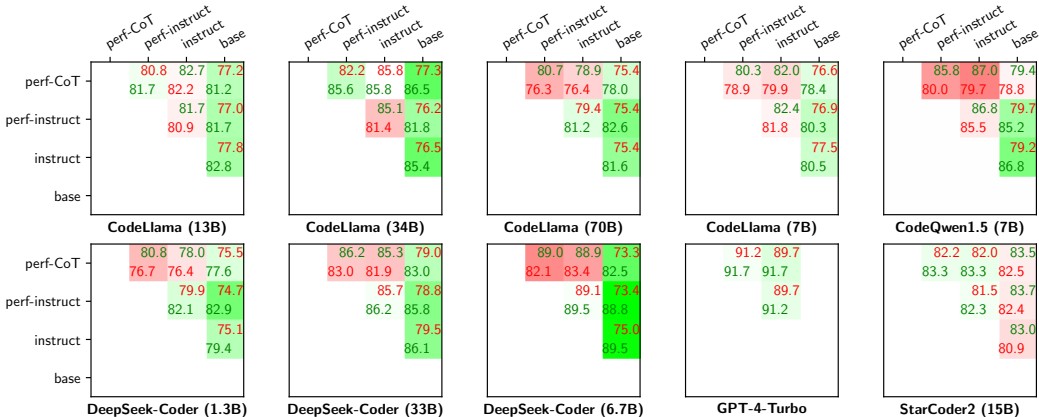

Figure 4: Pairwise comparison of DPS with model variant pairs. Each pair of variants is compared over the common set of passing solutions. Within each block, the bottom-left number comes from the corresponding variant in the y-axis and the top-right number is for the x-axis.

pairwise fashion and compute the efficiency scores over the common set of passing solutions to eliminate correctness inconsistency.

**Impact of instruction tuning.** Code instruction tuning finetunes the base model over high-quality code which can significantly improve the correctness in code generation (Wei et al., 2024; DeepSeek AI, 2023). Surprisingly, as is suggested by Figure 4, correct code generated by instruction-tuned models also tends to be more efficient than that of the base model (except for StarCoder2-15B). For example, instruction-tuned DeepSeekCoder-6.7B improves the base model by 19% regarding DPS. This interesting finding implies that general instruction tuning methods can improve multiple code quality aspects beyond correctness, even if the existing instruction tuning methods were not designed to optimize code efficiency.

**Impact of prompting.** For instruction-following models, besides the general chat template (*i.e.,* `instruct`), we also use two performance-encouraging zero-shot prompting methods (*i.e.,* `perf-instruct` and `perf-CoT`). Overall the performance-encouraging prompts neither consistently nor noticeably improve the code efficiency compared to using the basic prompting method. This shows that existing models are still weak in following such instructions, calling for future work to improve the instruction-following abilities of code LLMs. In the Appendix, Table 3 also shows that performance-encouraging prompts commonly lead to correctness degradation in code generation.

**Impact of model sizes.** It has been a general conclusion that larger models within a model family can often generate more accurate code. Figure 5 explores how parameter sizes within the same model family impact the overall code efficiency. Within the 12 pairs in Figure 5, there are seven cases where a larger model in the family outperforms the smaller one regarding code efficiency, *e.g.,* DeepSeekCoder-6.7B-

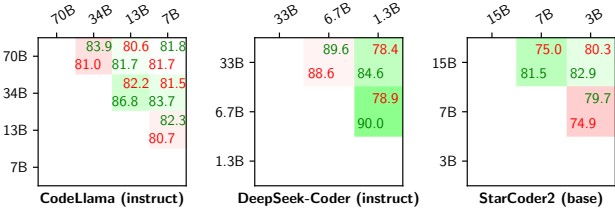

Figure 5: Pairwise DPS comparison with models of different parameter sizes.

Instruct improves the DPS of the 1.3B version by 14% and there is an 8.7% improvement from StarCoder2 7B to 15B. However, performance degradation with > 1% loss also happens 4 times, *e.g.,* in the worst case, there is a 6% degradation between the 3B and 7B versions of StarCoder2 base models. This underscores a new finding – the scaling law (Kaplan et al., 2020) persists for code correctness but does not seem explicit for code efficiency, calling for future research in modeling and data curation to improve efficiency in code generation models.

Lastly, we defer more evaluation details in Appendix A.3.

## 4.2 Evaluating the Evaluation

**SAS vs. EvalPlus.** EvalPlus (Liu et al., 2023a) generates an abundance of test cases, some of which could be already performance-exercising. Therefore, we use EvalPlus as a baseline to evaluate the performance difficulty of test inputs generated by SAS. Specifically, we use the filtering criteria in §2.3 as the evaluator and compare the number of retained tasks using the most challenging inputs of both methods, *i.e.,* more is better.

Table 1 shows the results. We start with a total of 563 tasks from Hu-manEval+ and MBPP+. Of these, for 342 tasks, we are able to obtain at least 10 validated solutions (§3). Using the criteria of "enough computation" which asks for test execution of over 10k instructions, the most performance-challenging EvalPlus inputs pass 204 out of 342 tasks (*i.e.,*

| | Total | Filtering | |
| --- | --- | --- | --- |
| | $(C \geq 10)$ | $> 10k$ instr. | $+ \# \text{Cluster} > 1$ |
| EvalPlus | 563 (342) | 204 | 25 |
| SAS (Ours) | | 271 | 121 |

Table 1: Retained tasks after different filtering phases using inputs from EvalPlus and SAS. $C \geq 10$ refers to tasks with at least 10 correct solutions from sampling.

60%), whereas SAS improves it by $1.3\times$ for enabling 271 tasks (*i.e.,* 79%) with inputs that exhibit larger computation. Meanwhile, since our cost measurement is based on hardware counters (§3), the variation-related criterion does not apply here. The clustering criterion allows tasks with at least 4 clusters of different performance characteristics. By applying this, EvalPlus-enabled tasks further reduce to 25 out of 204 (*i.e.,* 88% drop), whereas SAS stands out by passing 121 tasks, resulting in a relative improvement of $4.8\times$.

**Cross-platform variation.** A usable and reliable benchmark must easily run on various platforms and draw consistent conclusions. As such, we study the result consistency of EVALPERF over different test beds. Table 2 lists the DPS and DPS$_{\text{norm}}$ of three instruction-tuned models over *4* test-beds, covering a wide range of configurations covering desktop-, workstation-, and server-scenarios. With the emergence of heterogeneous CPUs, we also include a desktop using a heterogeneous architecture, *i.e.,* i9-12900K with 8 performance cores and 8 efficiency cores. All of these test beds are equipped with hardware counters (which are widely available), allowing for efficient profiling of #instructions. Specifically, Table 2 demonstrates that EVALPERF overall leads to very consistent conclusions, with a maximum coefficient of variation at 0.4%. Meanwhile, the evaluation takes approximately no more than 15 minutes for most evaluated models (*i.e.,* up to 10 profiled solutions for each of 121 tasks). This highlights the reliability and usability of the EVALPERF dataset as well as the DPE methodology. The low cross-platform variation comes from two major design choices: *(i) Differential evaluation:* in DPE the score is determined by the relative position compared against reference solutions which differentiate each other significantly; and *(ii) Hardware performance counters:* using hardware counters we can efficiently obtain reliable #instructions of profiled execution despite system noises.

| | | **Desktop** i7-10700K 8 Cores 64GB RAM | **Desktop** i9-12900K 8P & 8E Cores 64GB RAM | **Workstation** TR Pro 5975WX 32 Cores 256GB RAM | **Server** Xeon 6442Y 48 Cores 512GB RAM | **CV** (%) |
| --- | --- | --- | --- | --- | --- | --- |
| CODELLAMA-70B `instruct` | DPS | 79.2 | 79.4 | 79.4 | 78.8 | 0.3 |
| | DPS$_{\text{norm}}$ | 75.4 | 75.9 | 75.2 | 75.1 | 0.4 |
| DeepSeekCoder-33B `instruct` | DPS | 85.4 | 85.6 | 85.5 | 85.4 | 0.1 |
| | DPS$_{\text{norm}}$ | 78.6 | 78.5 | 78.6 | 78.4 | 0.1 |
| GPT-4 `instruct` | DPS | 90.5 | 90.0 | 90.7 | 89.9 | 0.4 |
| | DPS$_{\text{norm}}$ | 79.9 | 79.8 | 79.9 | 79.9 | 0.1 |

Table 2: Cross-platform variation of the mean Differential Performance Score.

## 5 Related Work

The correctness of general code generation is one of the most studied evaluation aspects. APPS (Hendrycks et al., 2021) and MBPP (Austin et al., 2021) curate Python problems with tests from coding websites. Meanwhile, HumanEval (Chen et al., 2021) includes 164 Python programming tasks manually designed from scratch. Yet, Liu et al. (2023a) shows that existing benchmarks have limited tests and proposes to extend the test coverage using automated test generation, creating HumanEval+ and MBPP+. Meanwhile, these Python tasks are translated to other languages for multilingual evaluation (Cassano et al., 2022; Athiwaratkun et al., 2022; Zheng et al., 2023). Furthermore, benchmarks also cover important domains such as data science (Lai et al., 2022; Yin et al., 2022; Zan et al., 2022), repository-level generation (Ding et al., 2023; Liu et al., 2023b), software development (Jimenez et al., 2023), security (Pearce et al., 2022), open-domain generation (Wang et al., 2023; Zhuo et al., 2024), and code understanding (Gu et al., 2024; Muennighoff et al., 2023; Liu et al., 2024). It is also important to avoid contamination in evaluation (Jain et al., 2024).

In terms of code efficiency, recent work PIE (Shypula et al., 2023) creates a benchmark to evaluate the program optimization capability of LLMs given base C++ programs. Furthermore, PIE (Shypula et al., 2023) employs CPU simulators to profile code execution to address the reproducibility concern. As a general evaluation mechanism, DPE (ours) as a meta technique can be applied to evaluating code generation and optimization, and also use CPU simulators for measurements. More concretely, our EVALPERF dataset based on DPE focuses on evaluating the code efficiency of the setting of direct code generation, which is more realistic in daily software development where oftentimes a base reference program is not available. Meanwhile, for measurement of computational cost, EVALPERF uses hardware performance counters that are low-overhead, reliable, and easy to use, explained in Appendix A.2.

At the time when the paper is accepted, there have been emerging sibling benchmarks for evaluating efficiency in code generation (Huang et al., 2024; Qiu et al., 2024; Waghjale et al., 2024). While these benchmarks consider additional source tasks (*e.g.,* those from LeetCode) and efficiency aspects (*e.g.,* memory usage), they still suffer from limitations including test computation insufficiency and relying on variation-sensitive compound metrics like average speedups. DPE as a meta-evaluation framework complements these new works and addresses these limitations by automatically creating performance-exercising test inputs and a stable compound metric mechanism for code efficiency. In addition, we suggest ablating impacts of the correctness dimension in efficiency evaluation, by comparing different models over a common set of passing tasks rather than the unaligned whole test suite (Huang et al., 2024).

## 6 Conclusion

This paper presents Differential Performance Evaluation (DPE), a novel framework to effectively assess the efficiency of code generated by Large Language Models (LLMs). By improving the computational complexity and metric mechanism of existing program synthesis benchmarks, DPE provides a general and robust approach to reasonably evaluate code efficiency. DPE includes two phases: *(i)* making a performance-exercising benchmark; and *(ii)* evaluating the global performance of new solutions. In the data curation phase, DPE transforms tasks into challenges that rigorously test code efficiency. In the evaluation phase, DPE profiles new solutions against reference solutions with representative performance characteristics. DPE is general, and based on it we create EVALPERF, a benchmark with 121 performance-challenging coding tasks.

Our evaluation based on EVALPERF reveals new insights into the impact of instruction tuning, promptings, and model sizes on code efficiency. Meanwhile, the evaluation of DPE itself showcases its effectiveness, reliability, and simplicity in performance benchmarking, even across various platforms.

## Acknowledgment

This work was partially supported by NSF grant CCF-2131943 and Kwai Inc, as well as API credits from the OpenAI Researcher Access Program.

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

# A Appendix

## A.1 Runtime Variation at Different Runtime Scales

We profile each validated solution sample from §3 5 times and draw the relation of its runtime variation and mean runtime in Figure 6, on the i7-10700K desktop (Table 2) without any other mandatory processes running. Specifically, the X-axis presents the mean runtime and the Y-axis presents the Coefficient of Variation (CV). Each blue dot in Figure 6 corresponds to a data point of one profiled solution and the purple bars draw the mean variation for each runtime magnitude at the scale of $10\times$. Figure 6 shows that in practice smaller executions are more sensitive to system noise despite using a clean test bed. This motivates the necessity of using large computations for robust and meaningful performance benchmarking.

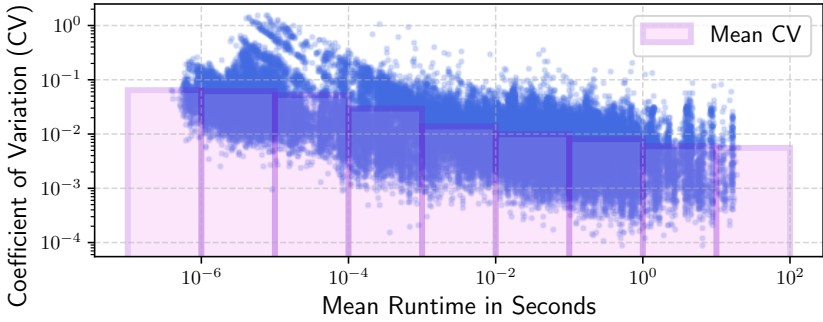

Figure 6: Distribution of runtime variation over the runtime scale.

## A.2 Discussion of Performance Measurement and Metrics

In this section, we discuss the pros and cons of different performance measurement methods as well as the commonly used metric of relative speedup.

**Measurements.** There are three primary measurements in the context of machine learning for code performance.

1. **Physical runtime** is the most commonly used metric for performance comparison (Mendis et al., 2019; Zheng et al., 2020; Zhou et al., 2020; Baghdadi et al., 2021; Mirhoseini et al., 2017). While physical runtime is easy to measure and can reflect physical differences, it can incur high variation in repetitive executions, especially in noisy environments (*e.g.,* shared cloud platforms that run various applications). As it is difficult to control the system noise when running performance benchmarking in the wild, it oftentimes challenges reproducibility.
2. **Architecture simulator** is used by Shypula et al. (2023) to address the cross-platform reproducibility issue. Architecture simulators such as Gem5 (Binkert et al., 2011) use software to simulate the CPU behaviors when executing a program and therefore its reproducibility is guaranteed by software. However, also because it is a software-based approach, simulating the program execution is oftentimes much slower than native execution and does not necessarily reflect the physical performance. Meanwhile, it is also not user-friendly to set up and run such simulators for interpreter-based programming languages such as the commonly used Python.
3. **Hardware performance counter** (Wikipedia, 2024) is used by EVALPERF (ours) to *natively* record the number of executed instructions between two program points. There are several benefits of using hardware counters: *(i) Usability:* performance counters are easy to use that one can simply query system calls between two program points to obtain the profile; *(ii) Efficiency:* performance counters only incur light-weight overheads (Weaver, 2013b); *(iii) Reproducibility:* the reported #instructions are highly reproducible that the observed variance of repeated executions are at most a few hundred instructions (*e.g.,* caused by context switch), whereas most computations in our benchmark can use billions of instructions. Note that #instructions might vary across platforms due to different

hardware and software settings. However, because EVALPERF follows DPE to determine performance ranking by referencing representative solutions, such inconsistency becomes invisible when computing DPS on different test beds, shown in Table 2. Last but not least, it does not mean that hardware counter can always replace simulators given that one of its major use cases is to mimic the behaviors of certain CPUs without having them physically.

Relative speedup is also a commonly used compound metric that is agnostic to the measurements mentioned above. Although straightforward for direct comparisons between two programs, its applicability becomes less clear across a broad range of tasks. This is because the degree of speedup varies significantly across different tasks, with the average speedup often skewed by tasks that allow for substantial improvements, such as optimizing an iterative solution over a recursive one for the n-th Fibonacci number calculation. This variability can lead to confusion, making it challenging to discern whether observed efficiency gains reflect a holistic improvement or are merely the result of optimizations for a narrow set of highly improvable tasks.

To overcome these limitations, Differential Performance Evaluation (DPE) defines a new metric, termed Differential Performance Score (DPS). Specifically, DPS for each task has a consistent range from 0 to 100% which stands for its empirical global performance position. This metric allows for an immediate understanding of a solution's effectiveness and the distance to optimal performance by comparison with benchmark solutions, providing a more nuanced and scalable approach to evaluating program efficiency improvements.

### A.3 Extended Results of Code Correctness and Efficiency

In this section, we complement §4.1 to show more result details in the code efficiency evaluation of LLMs. Specifically, Table 3 lists the detailed correctness on three sets of tasks and the performance scores of both DPS and $\text{DPS}_{\text{norm}}$ on EVALPERF. The sample-wise performance scores are consolidated by calculating the average, maximum, and minimum scores within the initial 10 samples of each task and then we aggregate the task-wise score by computing the average. Notably, different than the main evaluations in §4.1, efficiency scores in this section are computed over all passing tasks within each model for global comparison, *i.e.,* the correctness can impact the set of tasks that are used to compute the efficiency scores.

Moreover, Figure 8 offers a side-by-side comparison with Figure 7, illustrating $\text{DPS}_{\text{norm}}$ against DPS. Compared to DPS, $\text{DPS}_{\text{norm}}$ on EVALPERF tasks tend to be lower, indicating that the distribution of correct LLM samples is skewed to slower implementation. This is expected, as crafting efficient code often presents a greater challenge. Interestingly, while GPT-4 Turbo achieves the best DPS in Figure 7, using $\text{DPS}_{\text{norm}}$ the best-performing model becomes DeepSeekCoder-6.7B-instruct. This indicates that GPT-4 Turbo can generate code that is faster than the majority while DeepSeekCoder-6.7B-instruct tends to generate code that is on top of various efficiency levels.

Additionally, Figure 9 and Figure 10 show the correctness scores by computing pass@1 for the 121 EvalPerf tasks. Interestingly, we see a trend of reversed scaling law on CODELLAMA model family that smaller CODELLAMA models even achieve better pass rate. Despite this, other findings are overall consistent with that of Figure 7 and Figure 8.

| | **Correctness** (pass@1) | | | **DPS** | | | **DPS**$_{norm}$ | | |
|---|---|---|---|---|---|---|---|---|---|
| | HE+ | MBPP+ | EVALPERF | Avg | Max | Min | Avg | Max | Min |
| CODELLAMA-7B       base | **35.9** | **46.4** | 72.8 | 79.3 | **85.8** | 71.4 | 75.6 | **80.2** | 69.5 |
| instruct | **36.8** | 44.5 | 86.9 | **80.6** | 80.7 | **80.2** | **77.3** | 77.5 | **77.1** |
| perf-instruct | 33.0 | 44.1 | **90.5** | **80.1** | 80.2 | **79.8** | **77.3** | 77.3 | **77.0** |
| perf-CoT | 32.5 | 44.1 | **91.3** | 77.8 | 78.3 | 77.4 | 74.5 | 74.9 | 74.3 |
| CODELLAMA-13B      base | 38.7 | **50.4** | 76.7 | 75.6 | 80.6 | 67.7 | 74.5 | **77.2** | 70.0 |
| instruct | **40.2** | **50.6** | **88.9** | **80.3** | 80.7 | **78.7** | **75.8** | 76.3 | **75.2** |
| perf-instruct | 37.4 | 49.4 | 82.0 | **80.9** | **82.2** | **79.4** | **76.5** | **77.3** | **75.7** |
| perf-CoT | 36.4 | 49.1 | 85.9 | 78.7 | 80.9 | 75.5 | 75.5 | **76.8** | 73.5 |
| CODELLAMA-34B      base | 44.0 | **53.4** | 71.6 | 73.0 | 80.6 | 64.2 | 72.0 | 76.0 | 67.7 |
| instruct | **45.1** | **53.0** | **88.1** | **81.9** | 82.6 | **79.5** | 78.6 | 79.0 | 77.7 |
| perf-instruct | 43.3 | 50.7 | 87.1 | 79.8 | 82.0 | 78.3 | 78.5 | 79.2 | **77.8** |
| perf-CoT | 41.7 | 49.7 | 85.1 | **81.8** | **83.8** | **80.2** | **79.6** | **81.2** | **78.7** |
| CODELLAMA-70B      base | 50.2 | 52.9 | 75.5 | 75.9 | **84.8** | 65.8 | 73.8 | **79.3** | 68.6 |
| instruct | **62.5** | **60.2** | **81.7** | **79.2** | **85.4** | 71.2 | 75.4 | **79.3** | 70.1 |
| perf-instruct | 59.3 | 59.0 | 77.1 | **79.5** | 82.6 | **72.2** | 76.4 | 78.1 | **72.4** |
| perf-CoT | 60.2 | 58.6 | 78.8 | 76.9 | 82.2 | **71.3** | 74.6 | **78.6** | 70.8 |
| DeepSeekCoder-1.3B base | 25.6 | 45.9 | 74.3 | 74.7 | 79.4 | 69.6 | 74.6 | 78.0 | 71.1 |
| instruct | 59.0 | **50.7** | **84.5** | 80.0 | **83.6** | 76.2 | 75.5 | **77.9** | 73.9 |
| perf-instruct | **59.4** | 49.5 | **84.2** | **82.0** | **84.5** | **79.2** | **76.7** | **77.9** | **75.6** |
| perf-CoT | **60.3** | 47.0 | 75.4 | 77.1 | 82.6 | 69.5 | 74.1 | **77.7** | 69.9 |
| DeepSeekCoder-6.7B base | 36.3 | 55.2 | 71.3 | 73.3 | 80.9 | 64.2 | 70.6 | 74.8 | 65.9 |
| instruct | **74.9** | **63.2** | **92.9** | **89.9** | **90.7** | **88.5** | **81.4** | **81.6** | **80.9** |
| perf-instruct | **74.4** | 60.7 | 87.2 | 86.3 | 87.9 | 84.2 | 79.3 | 80.2 | 78.0 |
| perf-CoT | 70.5 | 59.9 | 83.4 | 81.8 | 88.5 | 74.5 | 78.1 | 82.0 | 74.6 |
| DeepSeekCoder-33B  base | 42.3 | 60.6 | 80.5 | 79.2 | 84.2 | 72.1 | 74.2 | 77.6 | 70.2 |
| instruct | **74.6** | **67.5** | **92.2** | **85.4** | **87.4** | 82.3 | **78.6** | 79.7 | 76.7 |
| perf-instruct | 71.6 | 66.5 | 85.3 | **86.3** | **87.9** | **84.3** | **79.1** | 80.3 | **77.9** |
| perf-CoT | 73.2 | 64.4 | 84.2 | 82.4 | 86.0 | 76.2 | **78.6** | **81.8** | 75.0 |
| StarCoder-3B       base | 17.1 | 35.4 | 59.3 | 80.3 | 86.1 | 71.5 | 77.9 | 82.3 | 72.4 |
| StarCoder-7B       base | 24.0 | 40.5 | 63.3 | 83.2 | 87.0 | 77.1 | 78.3 | 80.7 | 74.8 |
| StarCoder-15B      base | 28.7 | 48.1 | 66.7 | 80.4 | 85.0 | 76.3 | 75.6 | 78.3 | 73.4 |
| StarCoder2-3B      base | 26.1 | 45.7 | 69.4 | 80.5 | 84.1 | 76.5 | 77.6 | 80.1 | 74.9 |
| StarCoder2-7B      base | 29.4 | 45.4 | 68.3 | 73.8 | 81.6 | 65.3 | 73.6 | 77.9 | 69.5 |
| StarCoder2-15B     base | 37.8 | 54.2 | 74.0 | 82.1 | 88.2 | 73.8 | 77.2 | 80.7 | 73.2 |
| GPT-4 Turbo    instruct | 81.7 | **73.0** | **97.1** | 88.5 | 89.8 | 86.3 | 76.3 | 78.2 | 74.6 |
| perf-instruct | 82.3 | 71.2 | 94.9 | 90.5 | 92.2 | 87.5 | **79.9** | 81.3 | **77.4** |
| perf-CoT | **83.5** | 71.2 | **96.7** | **91.5** | **94.6** | **90.3** | **79.7** | **82.6** | **77.7** |

Table 3: Overall correctness and efficiency using 10 samples with temperature=0.2. Scores within one point of the best score per model are highlighted.

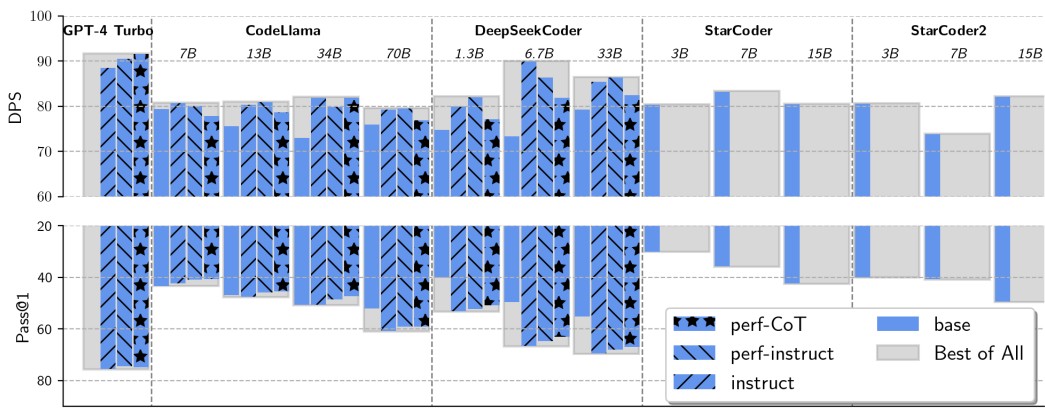

Figure 7: DPS on EVALPERF *v.s.* pass@1 on HumanEval+ and MBPP+.

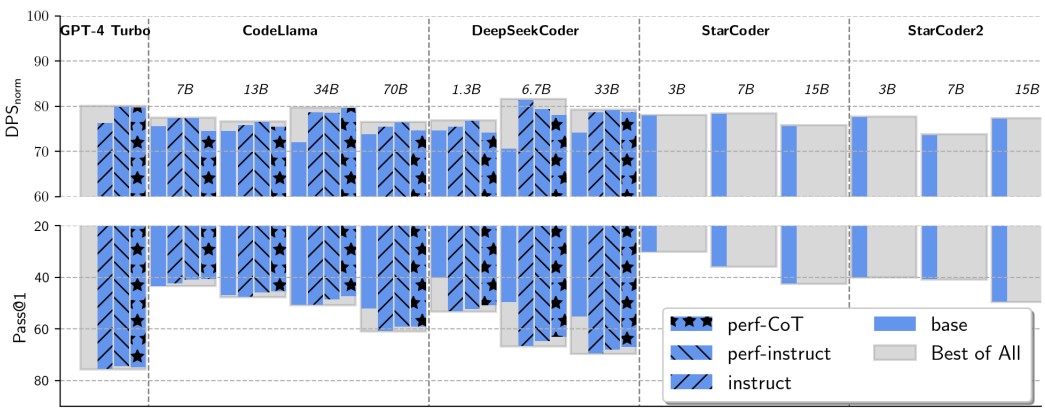

Figure 8: DPS$_{norm}$ on EVALPERF *v.s.* pass@1 on HumanEval+ and MBPP+.

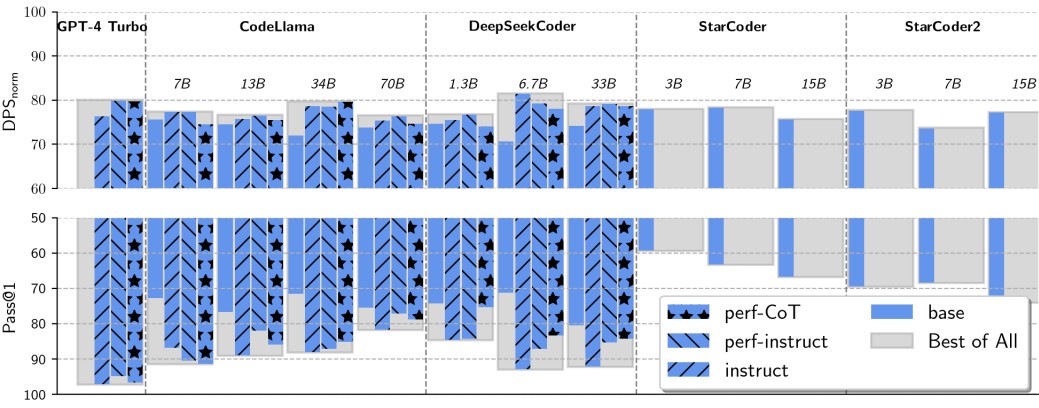

Figure 9: DPS on EVALPERF *v.s.* pass@1 on 121 EVALPERF tasks.

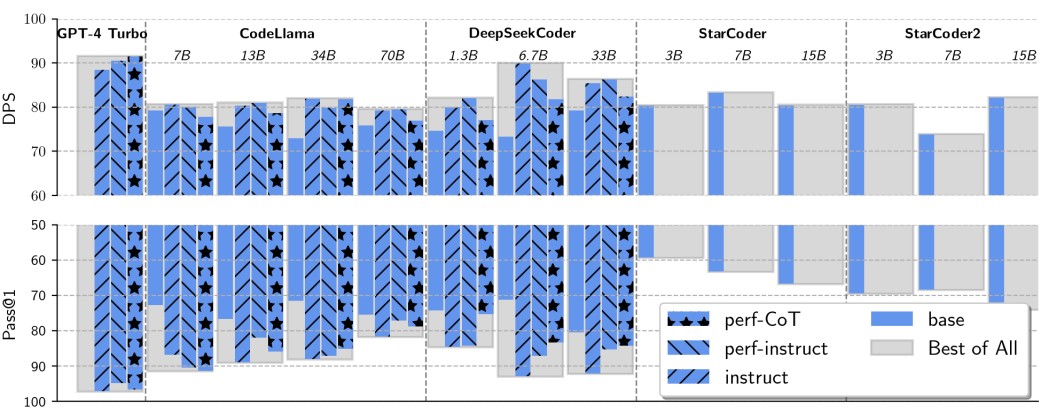

Figure 10: DPS$_{\text{norm}}$ on EVALPERF *v.s.* pass@1 on 121 EVALPERF tasks.

