# OpenReview forum: "Evaluating Language Models for Efficient Code Generation"
_colmweb.org/COLM/2024/Conference — COLM_

### Official Review · Reviewer_Jb72 · 2024-05-11

**Rating:** 6
**Confidence:** 4
**Ethics Flag:** 1

**Summary:**

This work proposes a methodology for evaluating efficiency of a given LLM-based
code generation technique (model + prompting). It takes a selection of Python
tasks from HumanEval and MBPP datasets along with a sample of reference
solutions from different LLMs, and builds a benchmark by clustering these
solutions on LLM-generated inputs. Crucially, the inputs to elicit different
complexity clusters are generated by task-specific _programs_, themselves
LLM-generated for every task. The performance score of a code-generation
technique is then the average percentile of performance of its generated
solutions among the clustered reference solutions for each task. The authors
evaluate five families of open-weight and proprietary LLMs and demonstrate that
high code correctness does not necessarily correlate with performance.

**Questions To Authors:**

# Minor

p2, line 1: "tp perform" => "performs"

**Reasons To Accept:**

# Strengths

* Important property to study and quantify.
* The proposed approach is general enough to apply across programming languages
  and architectures, provided a dataset of tasks to instrument.
* Exhaustive evaluation on many different LLMs, albeit with mixed results.
* Using LLMs to generate input-exercising programs instead of test inputs
  themselves is clever and novel. Similar techniques are used in programming
  competitions to create challenging inputs for contestants' solutions, but to
  my knowledge have not been used for exercising AI-generated code.

**Reasons To Reject:**

# Weaknesses and Questions

Most importantly, the proposed benchmark does not show a scaling signal, which
is often a red flag for its predictive power. When scaling the model (e.g.,
among the tiers of CodeLlama) one would expect the efficiency of generated code
to increase **if explicitly instructed to do so** (i.e., in "perf-prompted"
settings). However, it actually drops according to the paper's metric even as
code correctness increases - and that can be seen for almost all LLM families.
The paper provides no rationale or qualitative evaluation for these
observations, which in turn makes it difficult to attribute the misalignment to
the evaluated models or to the benchmark methodology itself.

I would expect some qualitative evaluation of examples to analyze when more
powerful models generate slower code. The simple tasks of HumanEval/MBPP make
such analysis possible.

The input-generating technique is useful and novel, thus should be evaluated as
an independent component.
1. Does the `scale` parameter actually correlate with "challenge" of the
   generated inputs? Do bigger values of `scale` lead to slower reference times
   across all tasks?
2. How do the methodology change if we simply asked for test generators
   parameterized with a random `seed`? Might it still elicit a useful
   distribution of challenging inputs?

---

> ### Author Rebuttal · Authors · 2024-05-31
>
> > Q1: ..examples to analyze when more powerful models generate slower code
>
> Thanks for the question! We case study some a few samples where DeepSeekCoder-6.7B-Instruct (perf-instruct) outperforms DeepSeekCoder-33B-Instruct (perf-instruct):
>
> Mbpp/732: Replace certain characters to ":".
> ```
> # The 6.7B model uses the simpler and faster "replace" function:
> def replace_specialchar(s):
>     return s.replace(' ', ':').replace(',', ':').replace('.', ':')
>
> # whereas the 33B model tend to use "re.sub" which is more powerful but slower as parsing regular expressions is costly:
> import re
> def replace_specialchar(s):
>     return re.sub(r'[ ,.]', ':', s)
> ```
>
> HumanEval/24: Finding largest divisor.
> ```
> # The 6.7B model smartly cut half of the computation in the beginning
> def largest_divisor(n: int) -> int:
>   i = n // 2
>   while i > 0:
>     if n % i == 0:
>       return i
>     i -= 1
>
> # The 33B model
> def largest_divisor(n: int) -> int:
>   for i in range(n-1, 0, -1):
>     if n % i == 0:
>       return i
> ```
>
> > Q2: ..Do bigger values of scale lead to slower reference times across all tasks?
>
> Thanks for the great question! We did see cases where input samplers produce "smaller"/"faster" inputs given a larger scale factor, e.g., sometimes GPT-4-turbo ignores the monotonicity requirement and simply generates random samplers. Notably, when making EvalPerf we add monotonicity assertions to falsify samplers that give "faster" inputs over larger scale factors.
>
> To study the frequency of monotonicity violation, we ran input samplers by setting the scale factor from 2 to 10 without monotonicity checking. Among the 563 initial tasks, samplers from 486 tasks can generate valid test inputs and 327 (67%) of them generate inputs that always respect computational monotonicity. We will add this in our revision.
>
> > Q3: ..if we simply asked for test generators parameterized with a random seed?
>
> Asking LLMs to produce random test generators has limitations in both theory and practice.
>
> In theory, when random sampling the tests, the scale/complexity of the test input is bounded by the range of randomness hardcoded by the LLM, e.g., the "1000" in `return [randint(0, 1000) for _ randint(0, 1000)]`. Such hardcoded values can still be too small to fully exercise the code solutions or too large to run. Meanwhile, for finding the largest computable input within a user-given time limit, searching inputs uniformly and randomly is less targeted and thus slower than searching inputs in monotonic scales.

---

### Official Review · Reviewer_TCT1 · 2024-05-11

**Rating:** 5
**Confidence:** 4
**Ethics Flag:** 1

**Summary:**

This paper proposes a new framework to evaluate the efficiency of code that large language models generate. I believe the authors' vision towards improving the efficiency of LLM-generated code is good and important: as LLMs generate more code, they should generate efficient code rather than slower code. In addition, the authors evaluate almost all the recent high-performing code LMs and give a very comprehensive evaluation of these models.

On the other hand, the domain of problems considered in this paper is relatively simple Python natural language to code programs. There are many more domains of programs that are optimized in real-world software engineering (such as code in different languages and performance-critical programs) that are not covered in this benchmark, which somewhat limits the scope of the work.

The paper, however, is relatively well-written and easy to follow. I would be more inclined to recommend acceptance of this work if it covered a wider variety of performance improvements and/or a more detailed analysis that revealed more insights of performance features of LLM generated code.

**Questions To Authors:**

- To me, there are two types of efficiency: algorithmic efficiency (e.g. O(n^2) to O(n) optimizations) and smaller, code-level optimizations (e.g. changing the order of for loops in matrix multiplication). The first is more focused from an algorithmic level, while the second is generally done after writing a piece of code and profiling it. Why do you decide to unify both of these under one benchmark?
- Can you more clearly differentiate the contributions of your benchmark from those of PIE (Shypula et al. 2023)? What insights do you believe your benchmark reveals that theirs does not?

**Reasons To Accept:**

- The paper focuses on a new angle in the evaluation of LLMs for code, namely program efficiency. Given that there are very few works looking at program efficiency, this benchmark could provide a valuable signal as more of these techniques appear.
- The authors do a comprehensive study of a variety of models consisting of different sizes, both open and closed source models, which provides information on the effectiveness of different classes of models in generating optimized code.
- The DPS technique is relatively general and can be applied to all sorts of coding problems.

**Reasons To Reject:**

- The size of EvalPerf is small, and there is no analysis of the variance of model scores on the benchmark. For example, it is hard to determine whether differences in the benchmark are statistically significant.
- While the paper uses prompt-based methods to try and evaluate the abilities of current language models, a more thorough attempt at asking models to generate efficient code would have been helpful.
- Scores on the benchmark are relatively high (e.g. in the 90s). The authors, however, note that DPS is a general technique that can be applied to other problems. However, the authors do not show any application of DPS to problems other than HumanEval and MBPP, which are relatively simple and already mostly saturated by the best models.
- There is no qualitative analysis

---

> ### Author Rebuttal · Authors · 2024-05-31
>
> > C1: ..there is no analysis of the variance..
>
> Please kindly note that Table 2 shows a negligible variation even when replicating the experiments over four different test beds.
>
> To study statistical significance, we rerun the experiment 40 times using the "instruct" and "perf-instruct" versions of DeepSeekCoder-6.7B (2 settings X 40 repeats X 50 samples X 121 tasks). Next, we use the Wilcoxon signed-rank test to check if the difference in the normalized DPS between the two settings is statistically significant, showing a P-value of 0.000379, which is much smaller than 0.05 (the most common significance level).
>
> > C2: ..application of DPS to problems other than HumanEval and MBPP..
>
> Please refer to C4 for reviewer 6zda.
>
> > C3: ...no qualitative analysis
>
> Please note that we did show a few qualitative conclusions, e.g., instruction tuning implicitly improves efficient code generating. One reason for limited qualitative analysis is that code efficiency is not optimized in training, thus we see more randomness than scaling laws, calling for research to optimize this direction. Nonetheless, Q1 for reviewer Jb72 includes some qualitative case studies.
>
> > Q1: Why ... unify both of these under one benchmark?
>
> EvalPerf evaluates end-to-end efficiency where both aspects can equally exhibit efficiency advantages.
>
> Why not put code-level efficiency under a program rewriting setting? Besides post-implementation optimizations, experienced developers can often directly write code of efficient paradigms. For example, in Python, it's faster to run built-in APIs like filter/map/reduce than vanilla loops. In C++, memory pre-allocation is a common efficiency tip where calling `vector::reserve` before `vector::push_back` reduces costly runtime allocations.
>
> > Q2: ..contributions of your benchmark from those of PIE
>
> PIE treats the efficiency problem in a **program repair** setting where a base solution is available, while DPE targets the **direct generation** of efficient code, as is exemplified in Q1.
>
> The unique contributions of DPE are to systematically create efficiency benchmarks and reliably evaluate them. As shown in "Limitation #1" (Section 1), prior work does not manage to use compute-costly tests and DPE specifically curates such tasks/tests to exhibit code efficiency. As shown in "Limitation #2", traditional speedup has unbound scales and we develop DPS to fix this issue. These can also be applied to the PIE setting. We will discuss more in our revision.

---

### Official Review · Reviewer_6zda · 2024-05-12

**Rating:** 7
**Confidence:** 4
**Ethics Flag:** 1

**Summary:**

This paper attempts to create a benchmark for performance evaluation of generated programs by utilizing existing sets of programs and seeing how well the model performs with respect to these. Overall, I definitely liked the paper and think that it could potentially be a good contribution to the conference.

**Questions To Authors:**

--- Small typo fix suggestions

which is pervasively available on most platforms
-> which is available on most platforms

the new solution (in the green box) matchs
-> the new solution (in the green box) matches

improving the the worst case scenario
-> improving the worst case scenario

There is a place that says "Compared to software profilers such as architecture simulators, hardware counters provides low overhead (Wikipedia, 2024)", but I would suggest a reference other than Wikipedia, as Wikipedia is generally a secondary source rather than a primary authoritative source.

There is a place where you say "It is worth noting that our scoring mechanism is agnostic to different efficiency measurements." in section 5. It is not clear what this means -- don't the results of your method vary depending on the architecture used for measurment (in section 4.2)? Maybe instead of "agnostic", "robust" would be a more appropriate word?

**Reasons To Accept:**

The paper is tackling a problem that has not gotten enough attention in the literature.

The "Synthesizing a Synthesizer" framework is clever, I like it!

Overall, the methodology and evaluation both are pretty solid. This was definitely the most well-done paper in my batch of reviews at COLM in this regard.

**Reasons To Reject:**

I still like this paper and think it should be accepted, but I did have a few concerns.

--- Major comments

One major concern that I had with the methodology, however, is that it seems that performance is only measured on programs where the answer is correct? If tht is the case, doesn't that introduce a major confounding factor that weaker models may generate answers to only the simpler programming problems, which are also easy to generate efficient solutions for? It seems that in order to rigorously compare two models with respect to their performance, the models would have to be compared across the same set of programs (so probably take the intersection of the sets of correct programs generated by the two models). I wonder if the authors

Another concern is that I found the methodology complex and pretty difficult to follow -- I hope that the clarity of the paper could be improved before it is finally published. Specifically, I found sections 2.4 and 2.5 pretty hard to follow due to their completxity and brief description. I was able to mostly understand them after a careful reading, but I would definitely encourage the authors to re-read these sections and try to make them clearer. Specific comments:
- Adding an illustrative example (e.g. to Figure 3) of the time1d, rdiff, and splitters values would be helpful to better understand.
- "mean execution time" is referenced in 2.4 before it is explained in 2.5, which made it confusing.
- Is "base" a typo for "bias"?
- Unless I missed it, it seems "cumulative ratio" is not defined anywhere? This is an important concept, and it was a bit hard for me to understand the methodology without knowing what this means.

--- Minor comments

But one possible issue with it is that it tries to generate inputs that would test the efficiency of the ground-truth solution (since the ground-truth solution is provided in the prompt). Programs other than the ground truth solution may be slow in different ways, and it is not guaranteed that this method would find inputs that exercise these other programs. Did you consider also generating programs that exercise the LLM-generated solutions instead?

One limitation of this work is that it only uses HumanEval and MBPP as inputs, which use the Python standard library only. It is not clear if this framework would scale up to testing more complex programs that rely on external libraries (like numpy or or PyTorch). It'd be good to at least mention this fact in the paper.

---

> ### Author Rebuttal · Authors · 2024-05-31
>
> > C1: ... models would have to be compared across the same set of programs...
>
> Thanks for the great suggestion! Indeed, it is important to compare model pairs over the same set of validated solutions. So we draw heatmaps in the following anonymous links:
>
> * Heatmap of $DPS$ comparison:  https://ibb.co/TBQvFV0
> * Heatmap of $DPS_{norm}$ comparison: https://ibb.co/5TkQbfY
>
> Specifically, we sort the models by their global scores. Each block in the figure shows the performance score of the Y-axis model and its improvement against the X-axis model (if the difference > 1pp). Overall, most higher-ranking models still outperform lower-ranking ones on the same set of solvable tasks. We will discuss more in our revision.
>
>
> > C2: ..hard to follow due to their complexity and brief description..
>
> Thanks for the suggestion! Indeed we should improve the methodology section by adding more explanation and examples. Regarding the detailed questions, "base" is a typo for "bias" and we should explain "cumulative ratio" in its definition. We will fix these in our revision!
>
>
> > C3: ..consider also generating programs that exercise the LLM-generated solutions
>
> Thanks for the question! We did study a handful of cases where we saw both ground truth and slower implementation can lead to similar test generators.
>
> Let's case-study HumanEval/39 which computes the n-th prime Fibonacci number. The ground truth tests primality using the Miller-Rabin algorithm with O(k logN), while LLMs can use a vanilla method with O(N) or O(sqrt(N)). Using our prompts, all of the three lead to the same input sampler:
>
> ```python
> def perf_input_gen(scale: int):
>   return (scale,)
> ```
>
> This might be reasonable as LLMs mainly use the reference solutions to ground code semantic understanding and their complexity might not lead to significantly different generators.
>
>
> > C4: ..testing more complex programs that rely on external libraries
>
> Thanks for the suggestions! We initially applied DPE on close-domain benchmarks as proof of concepts. We believe our technique can be applied to open-domain benchmarks such as DS-1000. We will include this in our future work.
>
> > Q1: ..results of your method vary depending on the architecture..
>
> Yes, hardware profilers are architecture-dependent. We were saying our "scoring mechanism", i.e., the metric of "differential performance score" (in S2.5), is agnostic to measurements (physical time, #instructions, etc.). We will fix the confusion in our revision.

---

> > ### Comment · Reviewer_6zda · 2024-06-04
> > **Thank you for the reply**
> >
> > Thank you for the reply, I appreciate the additional details and think adding them to the paper could be useful.

---

> > > ### Author Response · Authors · 2024-06-04
> > >
> > > Thank you for response! We will include these details. Let us know if you have more questions or concerns!

---

### Official Review · Reviewer_2P7H · 2024-05-18

**Rating:** 7
**Confidence:** 3
**Ethics Flag:** 1

**Summary:**

This paper introduces Differential Performance Evaluation (DPE), a framework for assessing the efficiency of code generated by LLMs. Unlike traditional benchmarks, DPE focuses on real-world efficiency-demanding tasks and provides a reliable metric for evaluation. By transforming coding tasks and creating the EVALPERF benchmark, the authors highlight insights into the impact of model sizes, instruction tuning, and prompting on code efficiency. The paper's evaluation underscores DPE's simplicity, speed, and cross-platform reliability, marking a significant advancement in code generation evaluation.

**Questions To Authors:**

n/a

**Reasons To Accept:**

1. The paper considers an interesting and important problem.

2. The paper proposes a novel framework for assessing the efficiency of code generated by LLMs.

3. The paper presents EVALPERF, including 121 performance-exercising programming tasks and test inputs.

**Reasons To Reject:**

1. The paper could potentially benefit from a more extensive comparison of EVALPERF Benchmarks across different open-source LLMs.

2. To enhance the paper, releasing the code used to reproduce the results would be beneficial.

---

> ### Author Rebuttal · Authors · 2024-05-31
>
> > C1: ... a more extensive comparison ... across different open-source LLMs.
>
> Thanks for the suggestion! For a more extensive comparison, in our rebuttal, we include evaluations of some recent open-source code models:
>
> * **CodeQwen1.5-7B**: DPS: 72.1 (%) / DPS_norm: 65.6 (%) / Pass@1: 68.2 (%)
> * **CodeQwen1.5-7B-Chat**:
>     * Instruct: DPS: 78.9 (%) / DPS_norm: 71.8 (%) / Pass@1: 78.8 (%)
>     * Perf-Instruct: DPS: 79.3 (%) / DPS_norm: 73.0 (%) / Pass@1: 75.2 (%)
>     * Perf-Cot: DPS: 74.2 (%) / DPS_norm: 69.3 (%) / Pass@1: 72.2 (%)
> * **starcoder2-15b-instruct-v0.1**:
>     * Instruct: DPS: 66.9 (%) / DPS_norm: 63.2 (%) / Pass@1: 71.9 (%)
>     * Perf-Instruct: DPS: 69.2 (%) / DPS_norm: 65.9 (%) / Pass@1: 72.5 (%)
>     * Perf-Cot: DPS: 67.5 (%) / DPS_norm: 62.8 (%) / Pass@1: 72.5 (%)
>
> Meanwhile, we include more experimental insights (e.g., pairwise comparison) for these models in C1 for Reviewer 6zda. In our next revision, we will try to include more if new code models are coming out.
>
>
> > C2: ... releasing the code used to reproduce the results would be beneficial.
>
> Thanks for the suggestion! We are enthusiastic about open research and have already open-sourced and packaged the code and the dataset. However, for review anonymity, we cannot disclose the pointers now but will include them in the final version.

---

### Author Response · Authors · 2024-05-31

We deeply appreciate all the reviewers for their insightful feedback and suggestions for our work!

In our responses below, we address each **primary question** (denoted as **Q**) or **comment** (denoted as **C**) raised by the individual reviewers. Additionally, we will revise our paper to incorporate editorial suggestions and add more comprehensive result analysis.

Should there be any misunderstandings of the questions, please kindly let us know. We are happy to communicate with all the reviewers throughout the discussion period.

---

### Decision · Program_Chairs · 2024-07-10

**Decision:**

Accept

**Comment:**

The paper introduces Differential Performance Evaluation (DPE), a novel framework for assessing the efficiency of code generated by large language models (LLMs). Unlike traditional benchmarks, DPE emphasizes real-world efficiency-demanding tasks, offering a reliable metric for evaluation. The authors transform coding tasks and develop the EVALPERF benchmark, comprising 121 performance-exercising programming tasks and test inputs. Their evaluation highlights the impact of model sizes, instruction tuning, and prompting on code efficiency, showcasing DPE's simplicity, speed, and cross-platform reliability.

This paper tackles a significant and underexplored problem in the evaluation of LLMs: the efficiency of generated code. The proposed Differential Performance Evaluation (DPE) framework and the accompanying EVALPERF benchmark are innovative contributions that offer valuable insights into code performance across various LLMs. The comprehensive evaluation, including recent high-performing models, underscores the potential of DPE to advance the field. However, several areas require improvement. Reviewer 2P7H and Reviewer TCT1 suggest expanding the benchmark to include a more diverse set of tasks and models, while Reviewer 6zda highlights the need for methodological clarity and rigorous comparison across models. Addressing these concerns, particularly by providing reproducibility through released code and enhancing methodological transparency, would strengthen the paper. Despite these limitations, the paper's contributions are significant, and with revisions, it holds promise for substantial impact in the field.